# Different selection practices affect the environmental sensitivity of beef cattle

**Anielly de Paula Freitas**[1]*, **Mário Luiz Santana Júnior**[2], **Flavio Schramm Schenkel**[3], **Maria Eugênia Zerlotti Mercadante**[2], **Joslaine Noely dos Santos Goncalves Cyrillo**[2], **Claudia Cristina Paro de Paz**[1,2]

**1** Department of Genetics, Ribeirão Preto Medical School, University of São Paulo, Ribeirão Preto, São Paulo, Brazil, **2** Advanced Beef Cattle Research Center, Animal Science Institute/APTA/SAA, Sertãozinho, São Paulo, Brazil, **3** Centre for Genetic Improvement of Livestock, Department of Animal Biosciences, University of Guelph, Guelph, Ontario, Canada

* aniellypf@hotmail.com

**Data Availability Statement:** All relevant data are within the paper and its Supporting Information files.

## Abstract

The objective of the present study was to evaluate the effects of different selection practices on the environmental sensitivity of reproductive and growth traits in males and females of three Nellore selection lines [control (NeC), selection (NeS), and traditional (NeT) lines]. Moreover, genetic trends for the intercept and slope were estimated for each line, and the possible reranking of sires was examined. A total of 8,757 records of selection weight (SW), 3,331 records of scrotal circumference (SC), and 2,311 records of days to first calving (DFC) from Nellore cattle born between 1981 and 2017 were analyzed. (Co)variance components and genetic parameters of all traits were estimated using a reaction norm model with Gibbs sampler. In all cattle lines, the mean heritability of the studied traits ranged from 0.39 to 0.75 for SW in both males and females, from 0.46 to 0.68 for SC, and from 0.06 to 0.57 for DFC along with the environmental descriptor. In all cattle lines, the genetic correlation coefficients between the intercept and slope ranged from 0.03 to 0.81 for SW, from -0.14 to 0.39 for SC, and from -0.87 to -0.42 for DFC. Genetic trends for the slope and proportion of plastic genotypes indicated that the NeS line was more responsive to environmental changes, whereas the NeC and NeT lines tended to respond more modestly. Reranking of sires was observed for all traits, specifically in the NeC and NeT lines, because of the weak correlation between the opposite extreme environments. In the NeS line, reranking of sires was observed for DFC alone. Our results indicate that the effects of genotype-environment interaction are important and should be considered in genetic evaluations of Nellore cattle. Moreover, different selection practices affected the environmental sensitivity of the Nellore selection lines tested in this study.

## Introduction

Phenotypic modifications induced by the environment are more common in quantitative traits of living organisms inhabiting heterogeneous environments [1]. The actions of

**Funding:** This work was supported by the São Paulo Research Foundation (FAPESP, Brazil, grant number #2016/17517-4 and #2019/01814-8), Coordenação de Aperfeiçoamento de Pessoal de Nível Superior – Brasil (CAPES Brazil, Finance Code 001) for the fellowship awarded to AdPF. This work was also supported in the form of grants by productivity research fellowship from National Counsel of Technological and Scientific Development (CNPq) awarded to CCPdP (#303972-2018-1), MLSJ (#301918/2017-1), and MEZM (#301918/2017-1). The funders had no role in study design, data collection and analysis, decision to publish, or preparation of the manuscript.

**Competing interests:** The authors have declared that no competing interests exist.

functional genes may be conditioned by a set of environmental factors, which can alter the genotypic and phenotypic traits as well as the genetic merit of animals in response to the environment they inhabit, ultimately changing the estimates of genetic parameters and ranking of animals for breeding [2–4]. Therefore, it is important to take into account the contribution of genotype-environment interaction (G×E) in genetic evaluations for increasing the efficiency of beef cattle selection programs, offering enhanced security in purchasing the most appropriate genetic material for a specific production environment.

When an organism produces a phenotypic response that varies as a continuous function of its environment, this relationship is called a reaction norm [5]. Reaction norm models (RNMs) can be developed for an infinite number of environments. Thus, infinite breeding values can be assigned for the same animal, and the phenotypic expression of a genotype along the environmental gradient can be assessed [6]. Previous studies have reported the high efficiency of RNMs for assessing the influence of G×E [7–9]. RNMs are effective since the animal response to selection can be predicted more accurately due to more reliable estimation of variance components as well as of direct and correlated responses at all time points along a trajectory, which is particularly useful when the phenotypes vary steadily along with the environmental descriptor.

In a study evaluating the effects of G×E, along with the estimation of parameters and genetic trends of growth traits in Nellore cattle, authors [10] reported that breeding values increased over time and confirmed the occurrence of G×E. These authors emphasized that sires with optimal breeding values in a particular region may not be the best in other regions. Similarly, the authors [7] reported genetic variation in the sensitivity of animals to different environments, highlighting the importance of selecting animals with genotypes that are more suitable for production in each environment.

Although several studies have reported the effects of G×E on growth and reproductive traits, genetic evaluations under breeding programs for beef cattle are performed under general assumptions of no G×E and constant residual and genetic variances. However, the effects of this interaction are evident, and improper modeling of these effects may lead to biased predictions of breeding values and consequently reduce genetic progress [11, 12], ultimately resulting in economic losses to the producer.

There have been few studies on the effects of selection on animals' responses to the environment. When comparing 100 most and 100 least heat-tolerant bulls and concluded that the former ones transmitted heat-tolerance to their daughters, which showed lower productive but higher reproductive performance [13]. Heat stress is a complex phenomenon triggering several response mechanisms in dairy cattle, and it negatively affects farm profitability because of antagonism between productivity and heat tolerance [14].

Therefore, the aims of the present study were to evaluate the effects of different selection practices on the environmental sensitivity of reproductive and growth traits of Nellore cattle lines using an RNM as well as to estimate genetic trends related to general production capacity (intercept of the reaction norm) and specific ability to respond to environmental variations (slope of the reaction standard) of each line.

## Materials and methods

### Ethics

Animal care and use committee approval was not required for this study because information was obtained from an existing database of the Advanced Beef Cattle Research Center of the Animal Science Institute, Sertãozinho, SP, Brazil.

## Data description

A total of 8,757 records of selection weight (SW) from males and females born between 1981 and 2017, 3,331 records of scrotal circumference (SC) collected at 378 days of age from males born between 1990 and 2017, and 2,311 records of days to first calving (DFC) from females born between 1981 and 2015 were analyzed.

The beef cattle selection program at this center began in 1976, with the objective of increasing the post-weaning weight of Nellore cattle based on individual performance, thus offering adequate data to assess the effects of selection practices on the environmental sensitivity of animals. In 1980, the Nellore population was divided into three selection lines. The control (NeC) line is a closed line in which sires from the same center were used, and the animals were selected for average post-weaning weight. The selection (NeS) line is another closed line, while the traditional (NeT) line is an open line in which sires from other populations both within and outside the same center were used, particularly during early years of the breeding program [15, 16]. In the NeS and NeT lines, the animals were selected for the highest differentials to increase post-weaning weight [15, 16]. Starting in 2012, the NeT line was selected for the highest differentials of post-weaning weight plus the lowest differentials of residual feed intake [17].

## Trait definitions

In the weight gain test initially performed at the Advanced Beef Cattle Research Center, the animals were weighed after fasting for food and water, and the weights were adjusted to values at 210 days of age based on the average daily gain from birth to weaning. After weaning, the males were confined in paddocks measuring 3,600 m$^2$, with food provided *ad libitum* twice a day. Feed included corn silage, hay, soybean meal, ground corn, and mineral salts with urea. During confinement, all males were subjected to a performance test in which they were weighed three times. The males were confined for 168 days; the first 56 days were considered the acclimatization period and were not accounted for in the analysis. After this period, weights adjusted for values at 378 days of age were obtained by adding the adjusted weaning weight to weight gain during the confinement period. After weaning, all females were sent to a pasture and weighed three to four times for a period of 340 days. After this period, the weight adjusted for value at 550 days of age was obtained by adding the adjusted weaning weight to weight gain during the 340-day period. The selection criteria used were weight adjusted to 378 days of age for males and weight adjusted to 550 days of age for females. In this study, these weights were considered the same trait (SW). In addition to SW, SC (cm) of males was measured at approximately 378 days of age, at the end of the performance test.

As described by [15], DFC data were obtained for all heifers that entered the breeding season, considering the difference between the dates of the beginning of the breeding season (November 15th to February 15th) and the subsequent calving. During quality control, data on artificial insemination, as well as on stillborn or twin calves, were excluded. Records from heifers that failed to calve were included, and a projected value was assigned to each mating record. The highest DFC record within each contemporary group (CG) by year and population of birth was identified, and 21 days were added to this record to generate the projected value for females that failed to calve [18].

## Environmental descriptor

Preliminary analysis of variance was performed using generalized linear modeling with SAS (SAS Institute Inc., Cary, NC, USA) to select non-genetic effects, such as birth year, selection line, birth month class, sex, sex–selection line interactions, and the sire effect for DFC alone, for inclusion in the model. To establish an environmental descriptor, selection line, sex, and

birth year for SW and selection line and birth year for SC were considered fixed effects in the definition of CGs for each trait. However, CG solutions corresponding to the NeC line alone were used, as they would more reliably describe the environmental conditions regardless of genetic trends since the beginning of the selection program. The CG for DFC included the birth year as the main effect, with records from animals of all selection lines, because the NeC line had no records for some studied years. CG solutions were obtained by fitting a standard animal model to the data of all studied animals, and CGs represented the environmental conditions the animals were subjected to over the years of selection [3, 19, 20].

## Data quality control

For quality control, CGs with fewer than ten animals; with the progeny of a single sire; with animals suffering from health problems; and without records of age, female age at calving, or weight at entry into the breeding season (for DFC) were excluded. Due to limited data for DFC, CGs with over four observations were maintained.

For SW, sex, and birth month class [August (1), September (2), October (3), and November–December (4)] were included as the fixed effects in the model. For SC, only the birth month class was included as the fixed effect in the model. For DFC, only the sire effect was included as the fixed effect in the model. The model also included the following covariables: age of the animal at measurement as a linear effect for all traits, age of dam at calving as linear and quadratic effects for SW and SC, and weight at entry into the breeding season for DFC. Table 1 summarizes the descriptive data of the three Nellore cattle selection lines obtained after quality control.

**Table 1. Descriptive data of the control (NeC), selection (NeS), and traditional (NeT) lines of Nellore cattle.**

| Variable | NeC | NeS | NeT |
|---|---|---|---|
| **Selection weight, kg** | | | |
| Animals in the pedigree | 1,964 | 3,756 | 4,975 |
| Sires | 86 | 138 | 180 |
| Dams | 437 | 899 | 1150 |
| Animals with measurements | 1,592 | 3,146 | 4,019 |
| Males | 820 | 1,562 | 1,979 |
| Females | 772 | 1,584 | 2,040 |
| Contemporary groups | 74 | 74 | 74 |
| Mean of the trait (standard deviation) | 264.7 (33.2) | 310.8 (48,7) | 314.6 (51.2) |
| **Scrotal circumference, cm** | | | |
| Animals in the pedigree | 1,204 | 2,468 | 3,159 |
| Sires | 66 | 114 | 146 |
| Dams | 276 | 566 | 692 |
| Males with measurements | 601 | 1,188 | 1,542 |
| Contemporary groups | 28 | 28 | 28 |
| Mean of the trait (standard deviation) | 21.9 (2.4) | 22.9 (2.5) | 23,6 (2.9) |
| **Days to first calving, days** | | | |
| Animals in the pedigree | 893 | 1469 | 1860 |
| Sires | 73 | 125 | 141 |
| Dams | 242 | 516 | 548 |
| Heifers with measurements | 450 | 926 | 935 |
| Contemporary groups | 35 | 35 | 35 |
| Mean age for entry into the breeding season (standard deviation) | 774.9 (23.2) | 769.8 (24.9) | 771.6 (24.9) |
| Mean of the trait (standard deviation) | 340.9 (35.6) | 349.4 (36.3) | 346.1 (35.7) |

## Data analysis

For data analysis, an RNM with homogeneous residuals was developed, in which the breeding value was expressed along with the environmental descriptor [21], and described as follows:

$$y_{ij} = fixed_i + \varphi_f \phi_f(CG_j) + a_j \phi_f(CG_j) + e_{ij} \qquad (1)$$

where $y_{ij}$ is the phenotype of animal $i$ in environment $j$; $fixed_i$ indicates the fixed effects (sex for SW, birth month class for SW and SC, and sire for DFC) and covariates [age of animal at measurement (linear) for all traits, age of dam at calving (linear and quadratic) for SW and SC, and weight at entry into the breeding season for DFC]; $\varphi_f$ is the fixed regression coefficient of $\Phi_f$; $\Phi_f$ is the second-order Legendre polynomial for CG solutions in environment $j$ (i.e., the environmental descriptor $CG_j$ obtained from the standard animal model; nested within sex for SW); $a_i$ is the random regression coefficient of the additive genetic effect of $CG_j$ on animal $i$; and $e_{ij}$ is the random residual associated with each animal $i$ along the environmental descriptor $CG_j$.

The additive genetic variances for a given environment X in the RNM were obtained using the following equation:

$$\sigma_a^2 | X = \sigma_a^2 + X^2 \sigma_b^2 + 2X \sigma_{a,b} \qquad (2)$$

where, $\sigma_a^2$ is the additive genetic variance of intercept of the reaction norm; $\sigma_b^2$ is the additive genetic variance of the slope of the reaction norm; and $\sigma_{a,b}$ is the covariance between the intercept and slope.

(Co)variance components for the selected traits were estimated by GIBBS2F90 [22] with Bayesian inference using Gibbs sampler and a single-trait animal model. The number of cycles was determined based on the trait and selection line studied. Analyses comprised single chains of 2,500,000 samples, with a burn-in period of 100,000 iterations and a thinning interval of 480 iterations, except for DFC in the NeT line for which single chains of 3,500,000 samples, with a burn-in period of 100,000 iterations and a thinning interval of 680 iterations, were used. Thus, variance components for the regression coefficients and genetic parameters were estimated from the remaining 5,000 samples.

Inferences for all (co)variance components and genetic parameters were based on mean, standard deviation, and 95% posterior probability. The posterior estimates were obtained using POSTGIBBSF90 [22]. Owing to the complexity of models and the small number of records available, several tests with a different number of chains and cycles were performed using the BOA package in R (R Core Team, R Foundation for Statistical Computing, Vienna, Austria, 2007) until convergence occurred for all parameters according to the Geweke criteria [23] and weak autocorrelation between the samples was attained.

## Genetic trends and environmental sensitivity

After estimating breeding values, genetic trends based on the average of estimates of the intercept (general production level) and slope (environmental sensitivity) of reaction norms for all traits according to the birth year were shown. A simple linear regression model was applied to these data to assess the significance of genetic trends. Finally, the possible effects of selection practices over the years on the general genetic merit and environmental sensitivity of cattle were evaluated.

## Phenotypic plasticity

An individuals' phenotypic plasticity was classified according to the standard deviation of slopes ($\sigma_b$) as an absolute value of $b_j$: $|b_j| < \sigma_b$ indicates robust genotypes, $\sigma_b \leq |b_j| < 2\sigma_b$

indicates plastic genotypes, and $|b_j| \geq 2\sigma_b$ indicates extremely plastic genotypes [19]. The ratio of variances of the slope and intercept was also calculated to infer about genetic variability associated with the environmental sensitivity of the reaction norm.

### Reranking of animals

To identify possible changes in ranking of animals, the top 25% sires of all selection lines with at least five progenies each were selected. The numbers of sires of the NeC, NeS, and NeT lines per trait, were 82, 135, and 164 for SW; 56, 94, and 117 for SC; and 51, 94, and 91 for DFC, respectively. Spearman rank correlations were computed to evaluate the possible changes in the ranking of sires under different environments (favorable, intermediate, and unfavorable) for each selection line.

The amplitudes of CG solutions representing the unfavorable, intermediate, and favorable environments, respectively, were 180, 206, and 250 kg for males and 115, 149, and 180 kg for females for SW; -1.59, 0.54, and 2.57 cm for SC; and 560, 518 and 484 days for DFC. These reflect the expected quality of environments the animals were subjected to over the selection years.

## Results

The posterior mean heritability of SW for males was higher than that for females of the NeC and NeS lines (NeC: 0.56 to 0.75 for males and 0.39 to 0.56 for females; NeS: 0.51 to 0.62 for males and 0.37 to 0.51 for females) but practically equal to that of the NeT line (0.53 to 0.65 for males and 0.53 to 0.64 for females). The estimates of heritability increased with improving environmental quality, except for females of the NeT line that showed decreasing heritability along with the environmental descriptor (Fig 1). These results indicate that response to selection may vary with environment and sex.

The posterior mean heritability of SC in opposite extreme environments was comparable between the NeC and NeT lines. However, estimates for the NeS line were slightly different. In the NeS line, the heritability estimates slightly increased with improving environmental quality; therefore, response to selection may vary with the environment (Fig 1). Mean heritability estimates along the environmental descriptor ranged from 0.55 to 0.66 for the NeC line, from 0.46 to 0.58 for NeS line, and from 0.56 to 0.68 for the NeT line.

The posterior mean heritability of DFC showed an increasing trend with improving environmental quality for all lines, with the NeS line showing the smallest increase (Fig 1). Mean heritability estimates along the environmental descriptor ranged from 0.26 to 0.57 for the NeC line, from 0.10 to 0.26 for the NeS line, and from 0.06 to 0.44 for the NeT line.

Genetic correlation coefficients between the intercept and slope of reaction norm for SW were 0.75 (0.14) and 0.81 (0.13) for the NeC and NeS lines, respectively, and null (0.03 ± 0.06) for the NeT line. This null correlation between the interception and slope of reaction norms of SW in the NeT line indicates possible reranking of animals under different environments; in this context, the selection of SW would result in little or no environmental sensitivity of animals. For SC, genetic correlation coefficients between the intercept and slope of reaction norms were 0.02 (0.22), 0.39 (0.33), and -0.14 (0.12) for the NeC, NeS, and NeT lines, respectively, which are considered low or even null due to the high standard deviation. These results also imply the possible reranking of animals under different environments, indicating that the best animals in a particular environment would not necessarily be the best in other environments. Genetic correlation coefficients between the intercept and slope of reaction norms for DFC were -0.49 (0.44), -0.42 (0.54), and -0.87 (0.17) for the NeC, NeS, and NeT lines, respectively. Similar to those of other traits, correlations of DFC for the NeC and NeS lines were

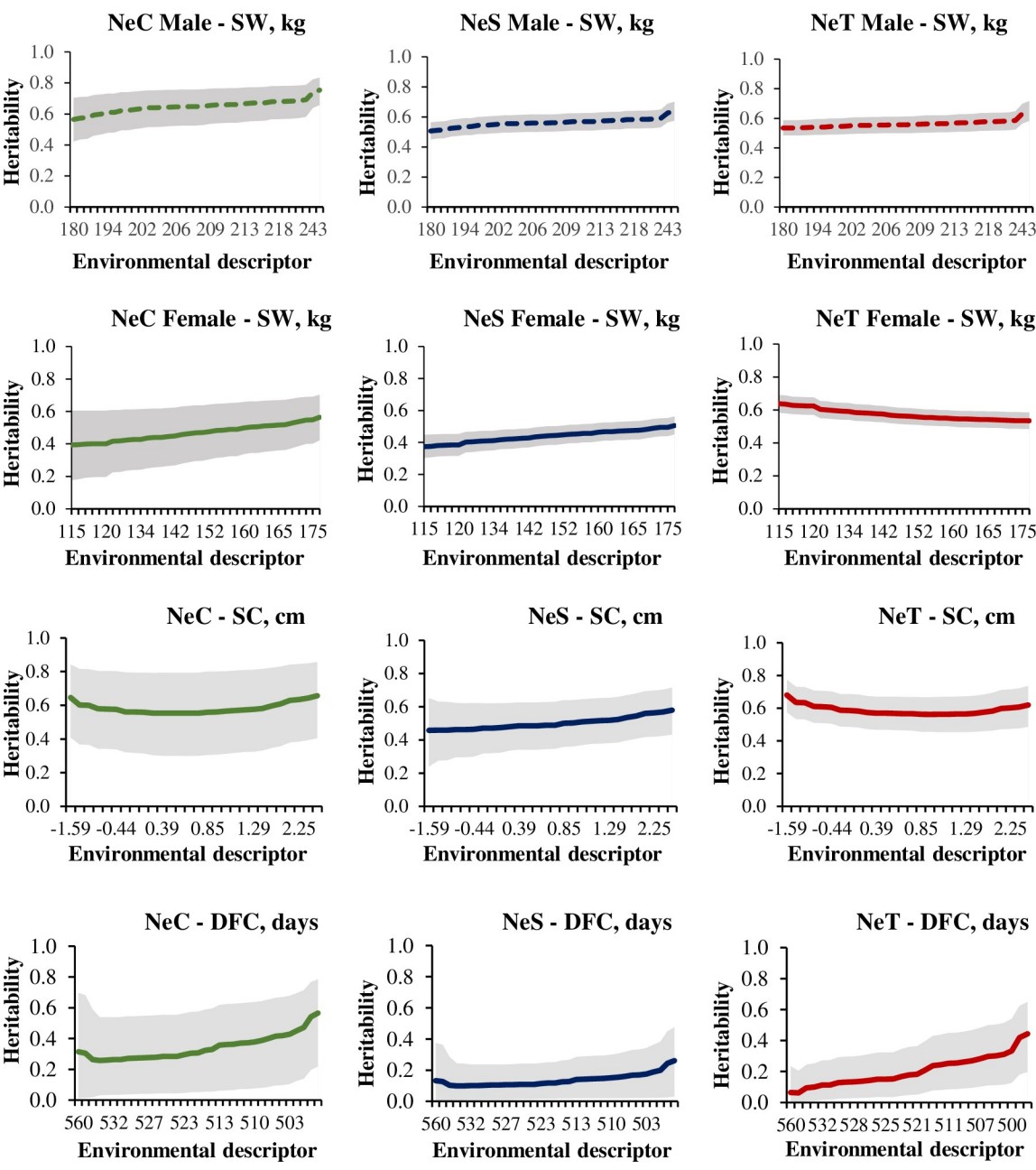

**Fig 1. Heritability estimates of Selection Weight (SW, kg) in males and females, Scrotal Circumference (SC, cm), and Days at First Calving (DFC, days) along the environmental descriptor for in three Nellore cattle lines (NeC, control; NeS, selection; and NeT, traditional line).**

considered null due to the high standard deviation, indicating possible reranking of animals in different environments.

Genetic correlation coefficients for SW in opposite extreme environments were high for all lines (0.95 for males and 0.80 for females of the NeC line; 0.98 for males and 0.95 for females of the NeS line; and 0.78 for males and 0.81 for females of the NeT line). Genetic correlation coefficients for SW between intermediate and extreme environments were also high for both males and females of all three lines, ranging from 0.92 to 1 under favorable environments and from

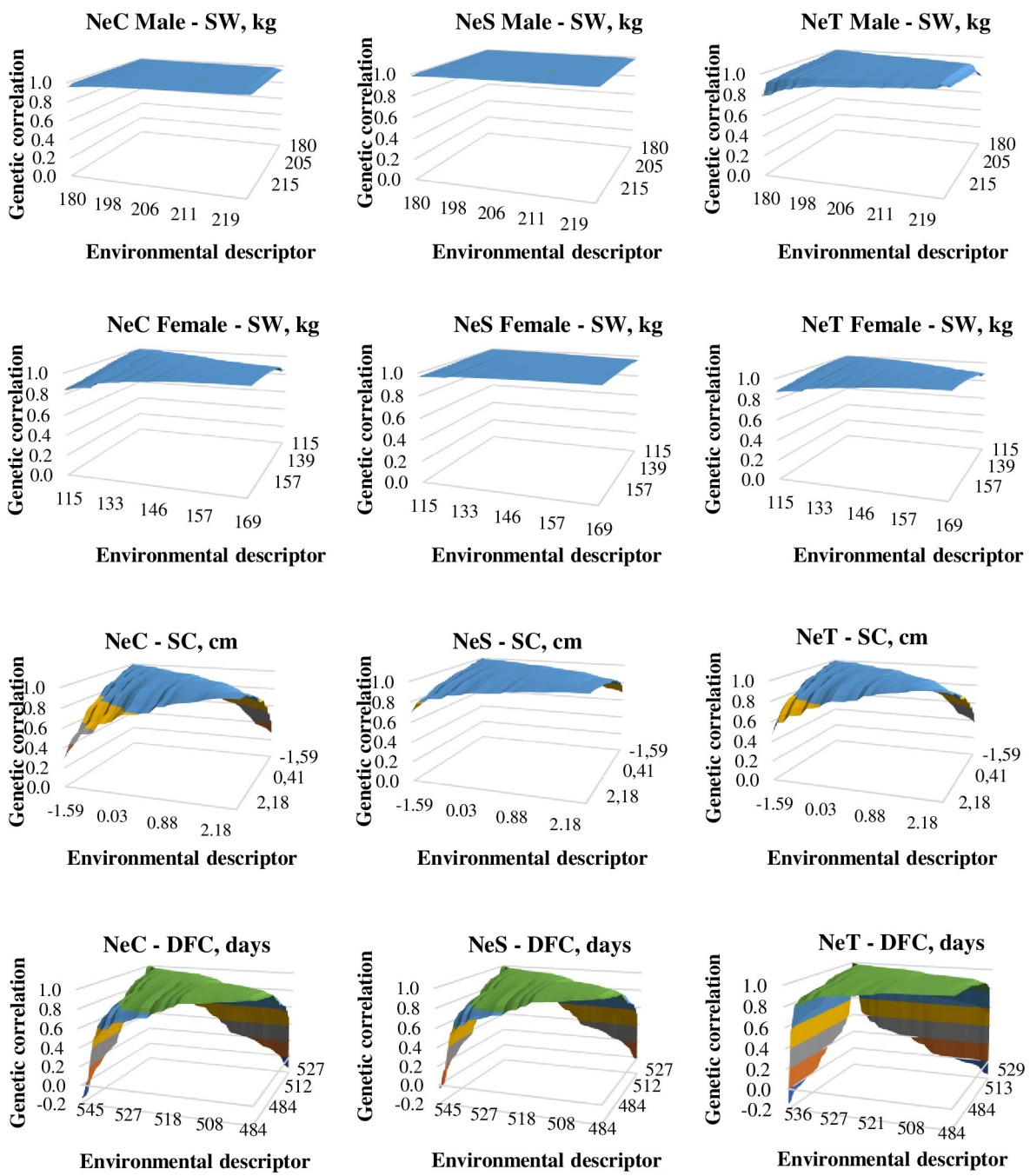

**Fig 2. Genetic correlations between Selection Weight (SW, kg) of males and females, Scrotal Circumference (SC, cm), and Days at First Calving (DFC, days) along the environmental descriptor in three Nellore cattle lines (NeC, control; NeS, selection; and NeT, traditional).**

0.93 to 0.99 under unfavorable environments (Fig 2). Genetic correlation coefficients for SC in opposite extreme environments were moderate for all lines (NeC, 0.28; NeS, 0.69; and NeT, 0.38) (Fig 2). Genetic correlation coefficients for SC between intermediate and extreme environments were also high (favorable: NeC, 0.81; NeS, 0.93; and NeT, 0.91; unfavorable: NeC, 0.79; NeS, 0.90, and NeT, 0.85). Contrary to those for other traits, genetic correlation

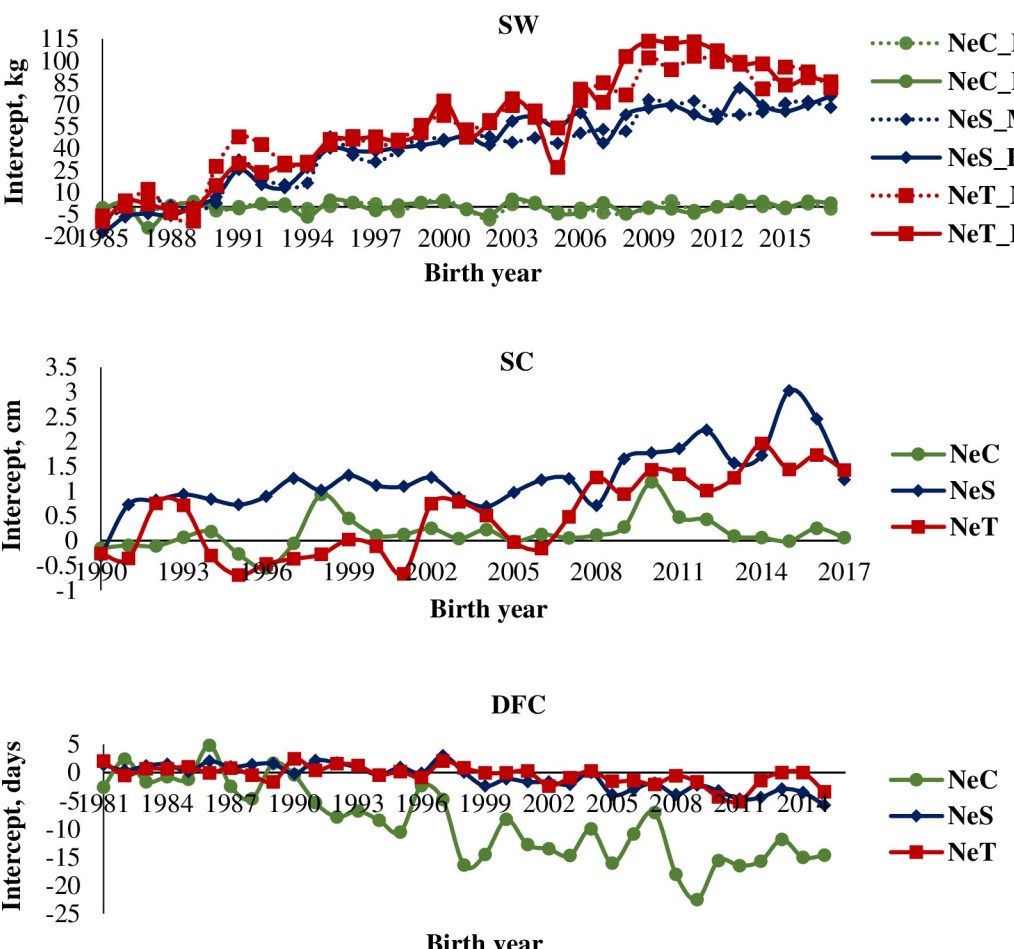

**Fig 3. Genetic trends for regression coefficients related to the intercept of reaction norms for Selection Weight (SW), Scrotal Circumference (SC), and Days at First Calving (DFC) in three Nellore cattle lines (NeC, control; NeS, selection; and NeT, traditional).** NeC_M, males of the control line; NeC_F, females of the control line; NeS_M, males of the selection line; NeS_F, females of the selection line; NeT_M, males of the traditional line; NeT_F, females of the traditional line.

coefficients for DFC in opposite extreme environments were low for all lines (NeC, -0.15; NeS, -0.04; and NeT, -0.15) (Fig 2). For all lines, genetic correlation coefficients for DFC between intermediate and favorable environments were high, ranging from 0.84 to 0.96. In contrast, those between intermediate and unfavorable environments were low, ranging from 0.12 to 0.50. These results suggest an important effect of G×E on DFC in different environments.

Regression coefficients of intercept of the reaction norm for SW were -0.01 (0.05) for males and 0.09 (0.06) for females of the NeC line; 2.60 (0.13) for males and 2.81 (0.14) for females of the NeS line; and 3.23 (0.18) for males and 3.52 (0.23) kg year$^{-1}$ for females of the NeT line (Fig 3). Genetic trend for the NeC line was non-significant (P > 0.05). The NeS and NeT lines tended to show higher productive performance for this trait, particularly the NeT line, which was expected since both lines were selected for the highest post-weaning weight differentials. Regression coefficients of intercept of the reaction norm for SC were 0.01 (0.01), 0.06 (0.01), and 0.08 (0.01) cm year$^{-1}$ for the NeC, NeS, and NeT lines, respectively (Fig 3). Genetic trend for the NeC line was non-significant (P > 0.05), while the NeS and NeT lines tended to show higher productive performance for this trait. Regression coefficients of intercept of the

reaction norm for DFC were decreasing and with values of -0.56 (0.06); -0.19 (0.02), and -0.12 (0.01) days year$^{-1}$ for the NeC, NeS, and NeT lines, respectively (Fig 3). Indicating higher reproductive performance of heifers, particularly of the NeC line.

Regression coefficients of slope of the reaction norm for SW were -0.003 (0.01) for males and 0.004 (0.02) for females of the NeC line; 0.39 (0.02) for males and 0.42 (0.02) for females of the NeS line; and -0.14 (0.06) for males and -0.13 (0.07) kg year$^{-1}$ for females of the NeT line (Fig 4). Genetic trends for both sexes of the NeC line as well as for females of the NeT line were non-significant (P > 0.05). Genetic trends for both sexes of the NeS line alone moved toward greater sensitivity, indicating that these animals are becoming more responsive to environmental changes. Regression coefficients of slope of the reaction norm for SC were -0.004 (0.001, 0.01 (0.001), and -0.01 (0.003) cm·year$^{-1}$ for the NeC, NeS, and NeT lines, respectively (Fig 4). Again, genetic trends for the NeS line alone moved toward a greater sensitivity, indicating a greater ability to respond to environmental changes. Regression coefficients of the slope of the reaction norm for DFC were 0.17 (0.02), 0.06 (0.01), and 0.06 (0.01) days·year$^{-1}$ for the NeC, NeS, and NeT lines, respectively (Fig 4), indicating that all three lines, albeit slowly, are moving toward greater sensitivity. Specifically, the NeC line presented the greatest increasing genetic trend for this trait.

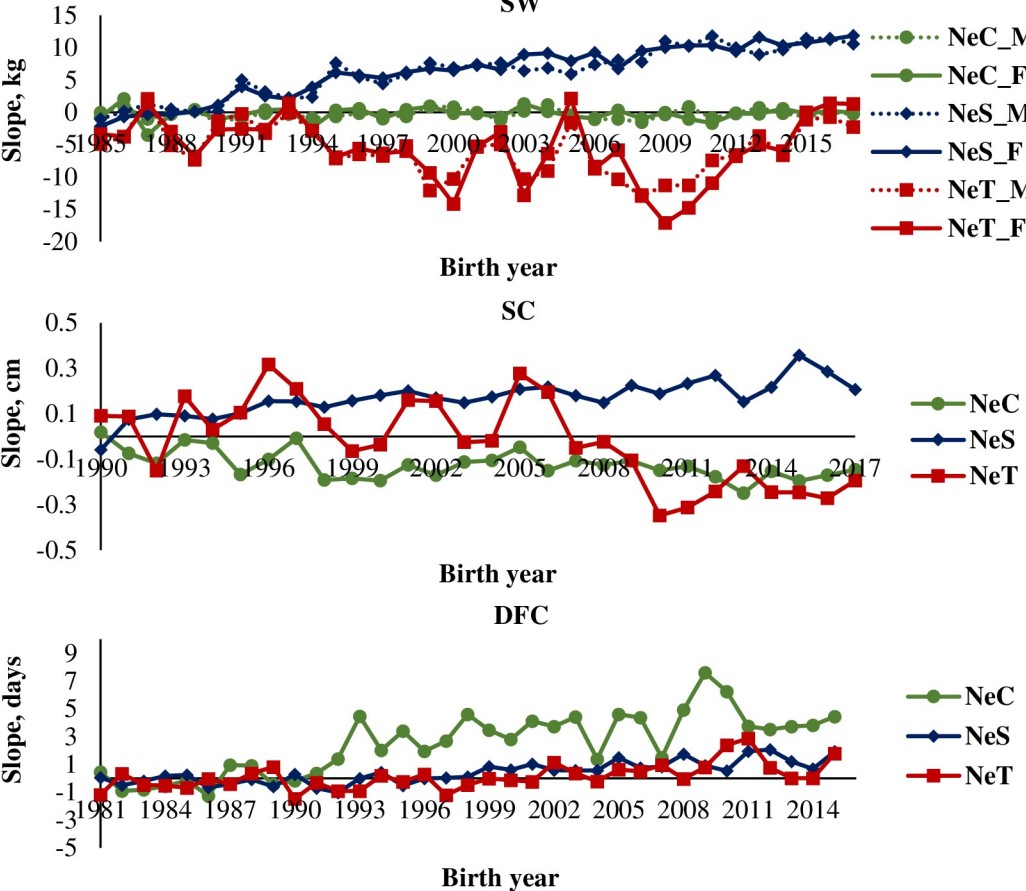

**Fig 4. Genetic trends for regression coefficients related to the slope of reaction norms for Selection Weight (SW), Scrotal Circumference (SC), and Days at First Calving (DFC) in three Nellore cattle lines (NeC, control; NeS, selection; and NeT, traditional).** NeC_M, males of the control line; NeC_F, females of the control line; NeS_M, males of the selection line; NeS_F, females of the selection line; NeT_M, males of the traditional line; NeT_F, females of the traditional line.

The observed frequencies of robust, plastic, and extremely plastic genotypes of SW, SC, and DFC are presented in Fig 5. Key differences in environmental sensitivity associated with SW could be verified, specifically in the NeS line, which was more sensitive to environmental changes. The proportions of plastic and extremely plastic genotypes were 30.98% in males and 30.83% in females of the NeC line, 54.80% in males and 56.31% in females of the NeS line, and 38.91% in males and 35.51% in females of the NeT line. The ratios of variances between the slope and intercept of the reaction norm for SW were 0.13, 0.04, and 0.20 for the NeC, NeS, and NeT lines, respectively, indicating that the NeS line showed the lowest genetic variability associated with environmental sensitivity proportional to the intercept. In contrast, the NeT line showed the greatest variability associated with environmental sensitivity.

The results for SC followed the trends for SW, indicating that key differences in environmental sensitivity associated with this trait could be verified, particularly for the NeS line, which showed more plastic genotypes than robust ones, corroborating the positive genetic trend for environmental sensitivity (slope). The percentages of plastic and extremely plastic genotypes were 34.28%, 51.26%, and 32.23% for the NeC, NeS, and NeT lines, respectively. The ratios of variance between the slope and intercept of reaction norm for SC were 0.19, 0.06, and 0.15 for the NeC, NeS, and NeT lines, respectively, indicating that the NeS line showed the lowest genetic variability associated with environmental sensitivity proportionally to the intercept. However, unlike that for SW, the NeC line showed the greatest genetic variability associated with environmental sensitivity for SC.

The ratios of variance between the slope and intercept of reaction norm for DFC were 0.44, 0.36, and 0.39 for the NeC, NeS, and NeT lines, respectively, indicating greater genetic variability associated with environmental sensitivity for DFC. The percentages of plastic and extremely plastic genotypes were 41.27%, 32.61%, and 32.62% for the NeC, NeS, and NeT lines, respectively. However, genetic trends based on regression coefficients of the slope of reaction norm for this trait indicated a high percentage of robust genotypes, that is, indicated a trend of increasing environmental sensitivity for these animals.

Spearman correlation analysis revealed potential changes in the rank of top 25% sires each selection line in each environment (Table 2). For SW, the Spearman correlation coefficients between favorable and intermediate environments were moderate for the NeC and NeT lines and high for the NeS line. In contrast, correlation coefficients between the favorable and unfavorable environments were non-significant for the NeC and NeT lines and high for the NeS line. For SC, Spearman correlation coefficients between favorable and intermediate environments were high for all lines, with the NeT line showing the lowest values among the three

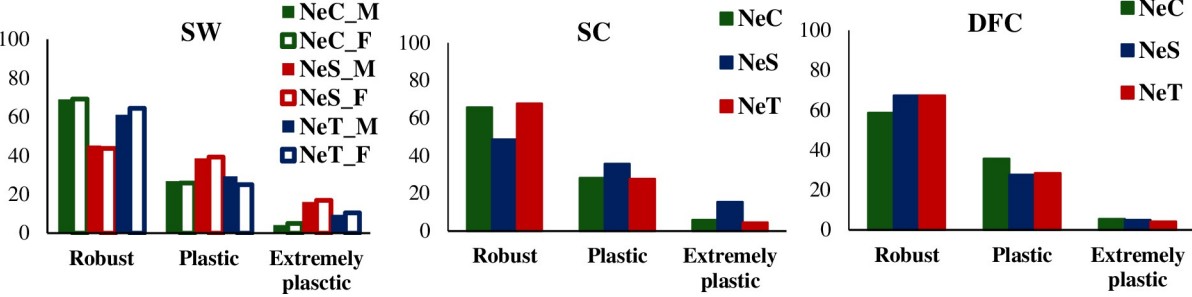

**Fig 5. Observed frequencies of robust, plastic, and extremely plastic genotypes for Selection Weight (SW), Scrotal Circumference (SC), and for Days at First Calving (DFC) in three Nellore cattle lines (NeC, control; NeS, selection; and NeT traditional).** NeC_M, males of the control line; NeC_F, females of the control line; NeS_M, males of the selection line; NeS_F, females of the selection line; NeT_M, males of the traditional line; NeT_F, females of the traditional line.

**Table 2. Spearman correlations of the top 25% sires according to breeding values of Selection Weight (SW), Scrotal Circumference (SC), and Days at First Calving (DFC) in control (NeC), selection (NeS), and traditional (NeT) lines of Nellore cattle obtained using the reaction norm model under different environments (favorable, intermediate, and unfavorable).**

| | NeC | | NeS | | NeT | |
|---|---|---|---|---|---|---|
| **SW** | | | | | | |
| | Unfavorable | Intermediate | Unfavorable | Intermediate | Unfavorable | Intermediate |
| Favorable | NS | 0.67[b] | 0.81[b] | 0.95[b] | NS | 0.41[b] |
| **SC** | | | | | | |
| | Unfavorable | Intermediate | Unfavorable | Intermediate | Unfavorable | Intermediate |
| Favorable | 0.56[a] | 0.83[b] | 0.87[b] | 0.96[b] | 0.46[a] | 0.78[b] |
| **DFC** | | | | | | |
| | Unfavorable | Intermediate | Unfavorable | Intermediate | Unfavorable | Intermediate |
| Favorable | NS | NS | 0.55[b] | 0.67[b] | NS | NS |

[a] P < 0.05

[b] P < 0.01, NS = non-significant.

lines. Spearman correlation coefficients between favorable and unfavorable environments were moderate for the NeC and NeT lines and high for the NeS line. Furthermore, for DFC, Spearman correlation coefficients between favorable and intermediate environments were non-significant for the NeC and NeT and moderate for the NeS line. Spearman correlation coefficients between favorable and unfavorable environments were also non-significant for the NeC and NeT and moderate for the NeS line.

## Discussion

In this study, the posterior mean heritability of SW in the NeC and NeS lines was higher in the most favorable environments, consistent with trends reported by [24] (heritability estimates, 0.28 to 0.55). In contrast, the NeT line showed a slight decrease in heritability, indicating that genetic parameters may vary depending on the environment.

Chiaia et al. [25] reported heritability estimates for SC ranging between 0.51 and 0.67, similar to the estimates obtained in this study; however, they reported that heritability increased as the environment became less restricted, which occurred only in the NeS line in the present study. For the other two lines, heritability estimates were similar under opposite extreme environments and lower in intermediate environments, corroborating the results of [26]. Therefore, we would expect a expressive response to the selection for SC based on the heritabilities.

The heritability estimates for DFC showed an increasing trend in all lines, with the highest values obtained in favorable environments. This trend indicates a possible greater response to selection in favorable environments, which would allow for the selection of more precocious heifers for breeding in these environments. Consistent with these trends, some studies [25, 27] observed that the heritability of reproductive traits of Nellore females increased as the environments became less restricted.

Genetic correlations between the intercept and slope of reaction norms for SW in the NeC and NeS lines indicated that a higher general productive performance of these lines is genetically associated with increased SW in better production environments. Although this trend is interesting, it may be detrimental in unfavorable environments, as animals may show poor performance. These trends corroborate the findings reported by [26]. In the NeT line, null correlation indicated possible reranking of animals in different environments, corroborating the findings of [24, 28]. Regarding reproductive traits of the NeC and NeS lines, the correlations

between the intercept and slope of the reaction norms were positive for SC but negative for DFC. However, due to high standard deviation, both traits showed trends indicating possible reranking of animals in different environments. In the NeT line, the correlations were for both traits. While the correlations for SC were practically null, those for DFC were high, indicating possible reranking of breeding values in different environments; these results are consistent with the reports of [26].

Genetic correlation coefficients below 0.80 [29] between the same traits in different environments demonstrate an important effect of G×E and indicate possible reranking of breeding values. Genetic correlations of SW in opposite extreme environments were high for all lines, except for males in the NeT line, indicating that this trait can respond to selection in any environment indicating that selection in any environment may be relevant for breeding programs aimed at selecting this trait under extreme environments, facilitating selection in different environments. In other words, the expression of SW is essentially the same across environments, as evidenced by [26].

All genetic correlation coefficients of SC in opposite extreme environments were below 0.80, demonstrating an important effect of G×E and indicating possible reranking of animals in these environments. Genetic correlations of SC in intermediate and opposite extreme environments were high in the NeS and NeT lines, corroborating the trends reported by [20]. Genetic correlation coefficients of DFC in opposite extreme environments were also below 0.80, and correlations in all lines were negative. Similarly, [25] reported negative genetic correlations of age at first calving (-0.27) in opposite environments, demonstrating an important effect of G×E in these environments.

Genetic trends observed for regression coefficients of the intercept of reaction norms for SW and SC were non-significant in the NeC line but significant and positive in the NeS and NeT lines, indicating genetic gain over the years for both lines selected for the highest differentials to increase post-weaning weight. Based on the greater genetic gain for SC, selection for the highest differentials to increase post-weaning weight may have a produced a positive effect on SC in the NeS and NeT lines.

All lines showed a genetic gain for DFC, albeit with negative trends, indicating that heifers tended to impregnate earlier during the breeding season, showing precocity, over the years; however, a less prominent downward trend was observed in the NeC line. This may be explained by the possible adverse effects of selection for the highest differentials to increase the post-weaning weight on DFC. In other words, selection to increase post-weaning weight likely impaired the reproductive performance of heifers of the NeS and NeT lines, at least compared with that of heifers of the NeC line (Fig 3).

Genetic trends for the regression coefficients of the slope of reaction norm for SW in the NeC line were non-significant; however, the NeS and NeT lines followed opposite trends. For animals of the NeS line, genetic trends for this trait indicated that this line is moving toward greater sensitivity, becoming more responsive to environmental changes. Moreover, animals of this line showed a significant genetic gain, as demonstrated by regression coefficients of intercept of reaction norm for this trait, which corroborates the trends reported by [20]. For animals of the NeT line indicated a decrease in the animals' sensitivity to environmental changes. Animals of this line also showed a significant genetic gain for SW, as demonstrated by the higher coefficient of intercept of reaction norm for this trait in the NeT line than in the NeS line.

Genetic trends observed for the regression coefficients of the slope of the reaction norm for SC were negative in the NeC and NeT lines and positive in the NeS line. As stated earlier, these opposite trends across the selected lines may be explained by the selection of the highest differentials to increase post-weaning weight in the NeS and NeT lines. The NeC and NeT lines

showed a decreased tendency to respond to environmental changes, corroborating the findings of [20]. Those authors reported a negative genetic trend of environmental sensitivity for SC in a study with Nellore cattle.

Genetic trends observed for the regression coefficients of the slope of reaction norm for DFC were positive in all lines, indicating that these lines, particularly the NeC line, are moving toward greater environmental sensitivity. Similarly, [30] observed high environmental sensitivity associated with DFC, indicating possible reranking of animals under different environments.

Phenotypic plasticity is the ability of organisms to alter their physiology or morphology in response to changing environmental conditions [31]. However, the effects of different selection practices on the environmental sensitivity of animals remain poorly understood. In a study by [32], the authors cited that the cost of phenotypic plasticity or environmental sensitivity, can mean a reduction in heritability and, consequently, in the response to selection. In the present study, the heritability estimates of SW and SC in the NeS line were lower than those in the NeT and NeC lines. In the NeS line, the animals are selected exclusively for the highest differentials to increase post-weaning weight, resulting in greater selection pressure on weight, which may have contributed to the reduction of genetic variability of this trait. In addition, as a consequence of the strong genetic correlation between weight and SC, a correlated response for SC has been observed [33]. The NeT line is selected for the highest differentials to increase post-weaning weight and RFI, which may contribute to reducing the selection pressure on weight in this line.

Phenotypic robustness and plasticity are closely related to G×E, and the presence of different patterns of reaction norms is indicative of the occurrence of G×E [34, 35], as evidenced in this study. We observed significant genetic trends associated with environmental sensitivity for all traits, specifically in the NeS line.

Regarding SW and SC, the NeC and NeT lines were considered more robust, with null or negative genetic trends, indicating that animals of these lines tend to respond to environmental changes more modestly; however, the performance of the NeT line was better than that of the NeC line and even better than that of the NeS line. Animals of the NeS line, meanwhile, were more sensitive to environmental changes, as evidenced by the higher frequency of plastic and extremely plastic genotypes (>50%) and the positive trend of coefficients of slopes of the reaction norms.

DFC was the most plastic trait, and all lines were considered plastic for this trait, with a greater genetic variability [34]. Despite the higher percentage of robust genotypes of this trait, all animals, particularly of the NeC line, tended to exhibit greater plasticity or environmental sensitivity. Based on the upward trends of coefficients of the slope of reaction norm for this trait and according to the results reported by [21], variation in the slope of reaction norm for a trait is directly associated with G×E, thus reflecting the environmental sensitivity of animals. In agreement with the results obtained in the present study, [32] reported that more plastic traits exhibited lower coefficients of heritability. In this sense, environmental sensitivity can be a determining factor for the response to selection.

An ideal breeding system would be the one in which genotypes show high performance with a slope close to zero, more robust animals with a better performance in different environments [36]. This could be the case for animals of the NeT line, which showed an overall higher performance than animals of the NeS line; however, these animals showed a greater percentage of robust phenotypes, with downward or null trends of coefficients of the slope of reaction norm for some traits.

Among the top 25% sires within a selection line, the correlation coefficients for traits were mostly below 0.80 in the NeC line, except for SC, which showed higher correlation coefficients

between favorable and intermediate environments. These results indicate reranking of animals of this line for all traits, confirming the importance of including G×E in genetic evaluations of these traits. The differences in correlation coefficients can be explained, at least in part, by three factors: first, differences in the environmental sensitivity of the animals, which respond differently according to the production environment. Second, differences in environmental sensitivity associated with heritability [32]. Third, the different selection practices to which the present population is subject.

Unlike those of the other lines, the rank of sires of the NeS line showed higher correlation coefficients than 0.80 for SW and SC, but lower correlation coefficients for DFC, indicating possible reranking of sires for DFC. Possible reranking for all traits was observed for sires of the NeT line. Similarly, [37] have reported weak correlations of traits between opposite environments.

The present study demonstrated an important effect of selection on the environmental sensitivity of animals. Genetic correlations between the traits of interest for selection, such as reproductive traits, can produce either positive or negative side effects. This is to be expected since according to the literature there is a strong genetic correlation between weight and SC [33] and a weak genetic correlation between weight and DFC [38], or perhaps because SC is not as plastic trait as reported by [20]. Nonetheless, the intriguing differences between the plastic NeS line and the robust NeT line may be due to many factors. For instance, the NeS line has been subjected to greater selection pressure, as it is selected only for the highest post-weaning weight differentials the line NeT is selected for RFI as well. Also, the NeT line was the only one to receive sires from both within and outside the Advanced Beef Cattle Research Center of the Animal Science Institute.

## Conclusions

Our results indicate that different selection practices interfered with the environmental sensitivity of the Nellore cattle lines tested in this study. The NeC line, but not the NeS and NeT lines, showed near null genetic trends for regression coefficients of the slopes of reaction norms for the selected traits. While the NeS line showed greater environmental sensitivity, the NeT line was more robust and less responsive to environmental changes. However, all lines showed a trend toward greater environmental sensitivity for DFC.

Our results also suggest that selection for the highest differentials to increase post-weaning weight affects the environmental sensitivity of animals, including their genetic parameters and ranking of breeding values, in addition to likely side effects on reproductive traits. We demonstrated the effects of G×E on all traits based on differences in heritability estimates along with the environmental descriptor, weak genetic correlations of the same traits between opposite extreme environments, and reranking of sires under different environments, particularly for the NeC and NeT lines and for DFC.

Our findings indicate that G×E is an important factor that should be accounted for in genetic evaluations. Environmental sensitivity of animals is an important trait that should be included in the selection indices for growth and reproductive traits of Nellore cattle.

## Supporting information

**S1 Table. Covariance component for Select Weight (SW), Scrotal Circumference (SC) and Days to First Calving (DFC) for each selection line given by reaction norm model.**
(DOCX)

## Author Contributions

**Conceptualization:** Anielly de Paula Freitas, Mário Luiz Santana Júnior, Flavio Schramm Schenkel, Maria Eugênia Zerlotti Mercadante, Claudia Cristina Paro de Paz.

**Data curation:** Anielly de Paula Freitas, Maria Eugênia Zerlotti Mercadante, Joslaine Noely dos Santos Goncalves Cyrillo, Claudia Cristina Paro de Paz.

**Formal analysis:** Anielly de Paula Freitas, Mário Luiz Santana Júnior, Flavio Schramm Schenkel, Maria Eugênia Zerlotti Mercadante, Claudia Cristina Paro de Paz.

**Funding acquisition:** Anielly de Paula Freitas, Claudia Cristina Paro de Paz.

**Investigation:** Anielly de Paula Freitas, Mário Luiz Santana Júnior, Flavio Schramm Schenkel, Claudia Cristina Paro de Paz.

**Methodology:** Anielly de Paula Freitas, Mário Luiz Santana Júnior, Flavio Schramm Schenkel, Claudia Cristina Paro de Paz.

**Project administration:** Claudia Cristina Paro de Paz.

**Resources:** Anielly de Paula Freitas, Claudia Cristina Paro de Paz.

**Software:** Anielly de Paula Freitas, Flavio Schramm Schenkel.

**Supervision:** Flavio Schramm Schenkel, Claudia Cristina Paro de Paz.

**Validation:** Anielly de Paula Freitas, Mário Luiz Santana Júnior, Flavio Schramm Schenkel, Maria Eugênia Zerlotti Mercadante, Claudia Cristina Paro de Paz.

**Visualization:** Anielly de Paula Freitas.

**Writing – original draft:** Anielly de Paula Freitas.

**Writing – review & editing:** Anielly de Paula Freitas, Mário Luiz Santana Júnior, Flavio Schramm Schenkel, Maria Eugênia Zerlotti Mercadante, Claudia Cristina Paro de Paz.

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
