## [Decision Letter · Decision Letter 0]

11 Jan 2021

PONE-D-20-29241

Different selection practices affect the environmental sensitivity of beef cattle

PLOS ONE

Dear Dr. de Paula Freitas,

Thank you for submitting your manuscript to PLOS ONE. After careful consideration, we feel that it has merit but does not fully meet PLOS ONE’s publication criteria as it currently stands. Therefore, we invite you to submit a revised version of the manuscript that addresses the points raised during the review process.

We look forward to receiving your revised manuscript.

Kind regards,

Raluca Mateescu

Academic Editor

PLOS ONE

Journal Requirements:

2. Please amend your list of authors on the manuscript to ensure that each author is linked to an affiliation. Authors’ affiliations should reflect the institution where the work was done (if authors moved subsequently, you can also list the new affiliation stating “current affiliation:….” as necessary).

Reviewers' comments:

Reviewer's Responses to Questions

**Comments to the Author**

1. Is the manuscript technically sound, and do the data support the conclusions?

Reviewer #1: Yes

Reviewer #2: Yes

2. Has the statistical analysis been performed appropriately and rigorously? 

Reviewer #1: Yes

Reviewer #2: Yes

3. Have the authors made all data underlying the findings in their manuscript fully available?

Reviewer #1: Yes

Reviewer #2: Yes

4. Is the manuscript presented in an intelligible fashion and written in standard English?

Reviewer #1: Yes

Reviewer #2: Yes

5. Review Comments to the Author

Reviewer #1: I compliment the authors on this work. It is a very good data set and an interesting project. I don't have concerns about the analysis or presentation of results. I do have some concern about possible over-interpretation of results. Some findings are certainly as expected, less GxE for SW than DFC, for example. We have found similar trends for growth traits versus stayability. I am not convinced that differences among the selection lines should be strong enough to explain the differences among the Spearman correlations presented in Table 2. These correlations vary quite markedly. How reliable are they? What are the biological explanations for these differences. Further, it would enhance the value of the paper to have greater discussion of the implications of the work. Even a few sentences would be helpful.

Reviewer #2: This manuscript addresses an important topic, is scientifically sound and well written. Below are some suggestions and a couple of errors that need to be corrected.

108 Should it read “Starting in 2012…”

109 More detail needs to be given on this selection scheme and how this altered the selection for weight

120 This should read :…by adding the adjusted weight…”

123 Same as above, add adjusted

126 This is confusing, "what is meant by SW of females was adjusted later"? Wasn't this already explained in the preceding sentences? If so, this sentence should be eliminated

131 Should this be "outside the expected calving season"? If so, how was that determined?

147 Should it be “…had no records for some studied years.”?

177 How would selection for weight impact this covariate?

229 Shouldn't these be reversed? Unfavorable CG should have the greatest number of days to first calving

252 Based on the Figure 1, I think these are the values for DFC and the values listed for DFC are for SC. Either the text is wrong or the figure is wrong.

258 A table with these correlations may be helpful.

369 For the other two traits you reported the percentage of plastic and extremely plastic; I recommend being consistent in reporting

407 This was not a response to selection, there was no selection for SC. We would expect a satisfactory response to selection for SC based on the heritabilities

486 There was little difference between the selection practices between NeT and NeS until late in the study. In the absence of extreme inbreeding along with extreme selection for the trait of interest, little reduction in genetic variation would be anticipated

527 This is to be expected since there is a strong genetic correlation between weight and SC and little genetic correlation between weight and DFC.

6. PLOS authors have the option to publish the peer review history of their article (what does this mean?). If published, this will include your full peer review and any attached files.

Reviewer #1: No

Reviewer #2: No

---

## [Author Response · Author response to Decision Letter 0]

17 Feb 2021

Reviewer's Responses to Questions

Reviewer #1: I compliment the authors on this work. It is a very good data set and an interesting project. I don't have concerns about the analysis or presentation of results. I do have some concern about possible over-interpretation of results. Some findings are certainly as expected, less GxE for SW than DFC, for example. We have found similar trends for growth traits versus stayability. I am not convinced that differences among the selection lines should be strong enough to explain the differences among the Spearman correlations presented in Table 2. These correlations vary quite markedly. How reliable are they? What are the biological explanations for these differences. Further, it would enhance the value of the paper to have greater discussion of the implications of the work. Even a few sentences would be helpful. 

R:The paragraph about Spearman correlation has been rewritten for better understanding (L375-384). We added two paragraphs to the discussion as suggested (L505-507 and 518-522).

Reviewer #2: This manuscript addresses an important topic, is scientifically sound and well written. Below are some suggestions and a couple of errors that need to be corrected.

108 Should it read “Starting in 2012…” 

R: It was corrected as indicated (L106).

109 More detail needs to be given on this selection scheme and how this altered the selection for weight. 

R: A reference was added to give details of NeT line. The inclusion of residual feed intake in the selection criterium of NeT line should not affect the weight selection since the genetic correlation between SW and RFI is low (Ceacero et al., 2016-PLosOne). However, the genetic gain in SW is expected to diminish since the animals have been selected for two traits after 2012 (L106-108).

- CEACERO TM, MERCADANTE MEZ, CYRILLO JNSG, CANESIN RC, BONILHA SFM, ALBUQUERQUE LG, HANSEN PJ. Phenotypic and Genetic Correlations of Feed Efficiency Traits with Growth and Carcass Traits in Nellore Cattle Selected for Postweaning Weight. Plos One, v. 11, p. e0161366, 2016.

120 This should read: …by adding the adjusted weight…” 

R: It was corrected as indicated (L119).

123 Same as above, add adjusted. 

R: It was corrected as indicated (L122).

126 This is confusing, "what is meant by SW of females was adjusted later"? Wasn't this already explained in the preceding sentences? If so, this sentence should be eliminated. 

R: Phrase eliminated as indicated.

131 Should this be "outside the expected calving season"? If so, how was that determined? 

R: Sorry, it was a mistake. The sentence was corrected according to the data edition (L129-130). 

147 Should it be “…had no records for some studied years.”? 

R: It was corrected as indicated (L146).

177 How would selection for weight impact this covariate? 

R: In the preliminary analysis of variance, it was found that the weight at entry into the breeding season was significant for DFC, since even though the heifers' age difference was significant, it was only 3 months.

229 Shouldn't these be reversed? Unfavorable CG should have the greatest number of days to first calving. 

R: You are right. It was corrected as indicated (L228).

252 Based on the Figure 1, I think these are the values for DFC and the values listed for DFC are for SC. Either the text is wrong or the figure is wrong. 

R: The text was wrong, but it has already been corrected (L250-251 and L254-255).

258 A table with these correlations may be helpful. 

R: We agree that the two forms of presentation are interesting. If this is not a requirement, we prefer to present the correlations in the text. OK?

369 For the other two traits you reported the percentage of plastic and extremely plastic; I recommend being consistent in reporting. 

R: It was changed as recommended (L367-369).

407 This was not a response to selection, there was no selection for SC. We would expect a satisfactory response to selection for SC based on the heritabilities. 

R: It was changed as recommended (L403-404).

486 There was little difference between the selection practices between NeT and NeS until late in the study. In the absence of extreme inbreeding along with extreme selection for the trait of interest, little reduction in genetic variation would be anticipated. 

R: The NeT line is selected for the highest differentials to increase postweaning weight and RFI, which may contribute to reducing the selection pressure on weight in this line (L483-487). 

527 This is to be expected since there is a strong genetic correlation between weight and SC and little genetic correlation between weight and DFC. 

R: The sentence was reformulated and references were added, since we did not analyze the correlations between these traits (L530-533).

Other text changes:

- All words "reclassified" have been replaced by "reranking".

---

## [Decision Letter · Decision Letter 1]

22 Feb 2021

Different selection practices affect the environmental sensitivity of beef cattle

PONE-D-20-29241R1

Dear Dr. de Paula Freitas,

We’re pleased to inform you that your manuscript has been judged scientifically suitable for publication and will be formally accepted for publication once it meets all outstanding technical requirements.

Kind regards,

Raluca Mateescu

Academic Editor

PLOS ONE

Additional Editor Comments (optional):

Reviewers' comments:

Reviewer's Responses to Questions

**Comments to the Author**

1. If the authors have adequately addressed your comments raised in a previous round of review and you feel that this manuscript is now acceptable for publication, you may indicate that here to bypass the “Comments to the Author” section, enter your conflict of interest statement in the “Confidential to Editor” section, and submit your "Accept" recommendation.

Reviewer #1: All comments have been addressed

Reviewer #2: All comments have been addressed

2. Is the manuscript technically sound, and do the data support the conclusions?

Reviewer #1: (No Response)

Reviewer #2: (No Response)

3. Has the statistical analysis been performed appropriately and rigorously? 

Reviewer #1: (No Response)

Reviewer #2: (No Response)

4. Have the authors made all data underlying the findings in their manuscript fully available?

Reviewer #1: (No Response)

Reviewer #2: (No Response)

5. Is the manuscript presented in an intelligible fashion and written in standard English?

Reviewer #1: (No Response)

Reviewer #2: (No Response)

6. Review Comments to the Author

Reviewer #1: (No Response)

Reviewer #2: (No Response)

7. PLOS authors have the option to publish the peer review history of their article (what does this mean?). If published, this will include your full peer review and any attached files.

Reviewer #1: No

Reviewer #2: No

---

## [Editor Report · Acceptance letter]

25 Mar 2021

PONE-D-20-29241R1 

Different selection practices affect the environmental sensitivity of beef cattle 

Dear Dr. de Paula Freitas:

I'm pleased to inform you that your manuscript has been deemed suitable for publication in PLOS ONE. Congratulations! Your manuscript is now with our production department. 

Kind regards, 

on behalf of

Dr. Raluca Mateescu 

Academic Editor

PLOS ONE